# Deep Graph Neural Networks
# via Flexible Subgraph Aggregation

## Abstract

Graph neural networks (GNNs), a type of neural network that can learn from graph-structured data and learn the representation of nodes through aggregating neighborhood information, have shown superior performance in various downstream tasks. However, it is known that the performance of GNNs degrades gradually as the number of layers increases. In this paper, we evaluate the expressive power of GNNs from the perspective of subgraph aggregation. We reveal the potential cause of performance degradation for traditional deep GNNs, i.e., aggregated subgraph overlap, and we theoretically illustrate the fact that previous residual-based GNNs exploit the aggregation results of 1 to $k$ hop subgraphs to improve the effectiveness. Further, we find that the utilization of different subgraphs by previous models is often inflexible. Based on this, we propose a sampling-based node-level residual module (SNR) that can achieve a more flexible utilization of different hops of subgraph aggregation by introducing node-level parameters sampled from a learnable distribution. Extensive experiments show that the performance of GNNs with our proposed SNR module outperform a comprehensive set of baselines.

## 1 Introduction

GNNs have emerged in recent years as the most powerful model for processing graph-structured data and have performed very well in various fields, such as social networks [1], recommender systems [2], and drug discovery [3]. Through the message-passing mechanism that propagates and aggregates representations of neighboring nodes, GNNs provide a general framework for learning information on graph structure.

Despite great success, according to previous studies [4, 5], GNNs show significant performance degradation as the number of layers increases, which makes GNNs not able to take full advantage of the multi-hop neighbor structure of nodes to obtain better node representations.

The main reason for this situation is now widely believed to be oversmoothing [4, 6, 5, 7]. However, since ResNet [8] uses residual connection to solve a similar problem in computer vision and obtains good results, several new works have been inspired to apply the idea of residual connection to GNNs to alleviate oversmoothing and thus improve the expressive power. For example, JKNet [5] learns node representations by aggregating the outputs of all previous layers at the last layer. DenseGCN [9] concatenates the results of the current layer and all previous layers as the node representations of this layer. APPNP [7] uses the initial residual connection to retain the initial feature information with probability $\alpha$, and utilizes the feature information aggregated at the current layer with probability $1 - \alpha$.

In this paper, we evaluate the expressive power of GNNs from the perspective of subgraph aggregation. Based on this perspective, we show that the single high-hop subgraph aggregation of message-passing

GNNs is limited by the fact that high-hop subgraphs are prone to information overlap, which makes the node representations obtained from k-hop subgraph aggregation indistinguishable, i.e., oversmoothing occurs.

Based on this perspective, we conduct a theoretical analysis of previous residual-based models and find that previous methods are in fact able to utilize multiple subgraph aggregations to improve the expressiveness of the model. However, most methods tend to utilize subgraph information by fixed coefficients, which assumes that the information from the subgraph of the same hop are equally important for different nodes, which leads to inflexibility in the model's exploitation of subgraph information and thus limits further improvement of the expressive power. Some existing methods try to overcome this inflexibility but lead to overfitting by introducing more parameters, which in turn affects the effectiveness of the model, which is demonstrated by the experiment.

Considering these limitations, we propose a **S**ampling-based **N**ode-level **R**esidual module (**SNR**). Specifically, we adopt a more fine-grained node-level residual module to achieve a more flexible exploitation of subgraph aggregation, which is proved by the theoretical analysis. On the other hand, to avoid overfitting due to the introduction of more parameters, instead of learning the specific parameters directly, we first learn a correlation distribution through reparameterization trick and obtain the specific residual coefficients by sampling. Experiments verify that this sampling-based approach can significantly alleviate overfitting.

**Our Contributions.** (1) We reinterpret the phenomenon that the effectiveness of traditional message-passing GNNs decreases as the number of layers increases from the perspective of $k$-hop subgraph overlap. (2) Based on the idea of subgraph aggregation, we theoretically analyze the previous residual-based methods and find that they actually utilize multiple hop subgraph aggregation in different ways to improve the expressive power of the model, and we point out the limitations of inflexibility and overfitting in previous residual-based methods. (3) We propose a sampling-based node-level residual module that allows more flexible exploitation of different $k$-hop subgraph aggregations while alleviating overfitting due to more parameters. (4) Extensive experiments show that GNNs with the proposed SNR module achieve better performance than other methods, as well as with higher training efficiency, on semi-supervised tasks as well as on tasks requiring deep GNNs.

## 2  Preliminaries

### 2.1  Notations

A connected undirected graph is represented by $\mathcal{G} = (\mathcal{V}, \mathcal{E})$, where $\mathcal{V} = \{v_1, v_2, \ldots, v_N\}$ is the set of $N$ nodes and $\mathcal{E} \subseteq \mathcal{V} \times \mathcal{V}$ is the set of edges. The feature of nodes is given in matrix $\mathbf{H} \in \mathbb{R}^{N \times d}$ where $d$ indicates the length of feature. Let $\mathbf{A} \in \{0, 1\}^{N \times N}$ denotes the adjacency matrix and $\mathbf{A}_{ij} = 1$ only if an edge exists between nodes $v_i$ and $v_j$. $\mathbf{D} \in \mathbb{R}^{N \times N}$ is the diagonal degree matrix whose elements $d_i$ computes the number of edges connected to node $v_i$. $\tilde{\mathbf{A}} = \mathbf{A} + \mathbf{I}$ is the adjacency matrix with self loop and $\tilde{\mathbf{D}} = \mathbf{D} + \mathbf{I}$.

### 2.2  Graph Neural Networks

A GNNs layer updates the representation of each node via aggregating itself and its neighbors' representations. Specifically, a layer's output $\mathbf{H}'$ consists of new representations $\mathbf{h}'$ of each node computed as:

$$\mathbf{h}'_i = \mathbf{f}_\theta \left( \mathbf{h}_i, \textbf{AGGREGATE} \left( \{ \mathbf{h}_j \mid v_j \in \mathcal{V}, (v_i, v_j) \in \mathcal{E} \} \right) \right)$$

where $\mathbf{h}'_i$ indicates the new representation of node $v_i$ and $\mathbf{f}_\theta$ denotes the update function. The key to the performance of different GNNs is in the design of the $\mathbf{f}_\theta$ and **AGGREGATE** function. Graph Convolutional Network (GCN)[10] is a classical massage-passing GNNs follows layer-wise propagation rule:

$$\mathbf{H}_{k+1} = \sigma \left( \tilde{\mathbf{D}}^{-\frac{1}{2}} \tilde{\mathbf{A}} \tilde{\mathbf{D}}^{-\frac{1}{2}} \mathbf{H}_k \mathbf{W}_k \right) \tag{1}$$

where $\mathbf{H}_k$ is the feature matrix of the $k^{\text{th}}$ layer, $\mathbf{W}_k$ is a layer-specific learnable weight matrix, $\sigma(\cdot)$ denotes an activation function.

## 2.3 Residual Connection

Several works have used residual connection to solve the problem of oversmoothing. Common residual connection for GNNs are summarized below. Details are explained in Appendix A.

Table 1: Common residual connection for GNNs.

| Residual Connection | Corresponding GCN | Formula |
|:---:|:---:|:---:|
| Res | ResGCN | $\mathbf{H}_k = \mathbf{H}_{k-1} + \sigma\left(\tilde{\mathbf{D}}^{-\frac{1}{2}}\tilde{\mathbf{A}}\tilde{\mathbf{D}}^{-\frac{1}{2}}\mathbf{H}_{k-1}\mathbf{W}_{k-1}\right)$ |
| InitialRes | APPNP | $\mathbf{H}_k = (1-\alpha)\tilde{\mathbf{D}}^{-\frac{1}{2}}\tilde{\mathbf{A}}\tilde{\mathbf{D}}^{-\frac{1}{2}}\mathbf{H}_{k-1} + \alpha\mathbf{H}$ |
| Dense | DenseGCN | $\mathbf{H}_k = \mathbf{AGG}_{dense}(\mathbf{H}, \mathbf{H}_1, \ldots, \mathbf{H}_{k-1})$ |
| JK | JKNet | $\mathbf{H}_{output} = \mathbf{AGG}_{jk}(\mathbf{H}_1, \ldots, \mathbf{H}_{k-1})$ |

## 3 Motivation

Message-passing GNNs recursively update the features of each node by aggregating information from its neighbors, allowing them to capture both the graph topology and node features. For a message-passing GNNs without a residual structure, the information domain of each node after $k$-layer aggregation is a related $k$-hop subgraph. Figure 1 shows that, after two aggregation operations, nodes on layer 2 obtain 1-hop neighbor and 2-hop neighbor information in layer 0, respectively. According to the definition of the $k$-hop subgraph, the information of the node on layer 2 in the figure is composed of all reachable nodes information shown on layer 0. We can consider the result of $k$-layer residual-free message-passing GNNs is equivalent to $k$-time aggregation of each node on its $k$-hop subgraph, which we call $k$-hop subgraph aggregation.

It is evident that as the number of aggregation operations increases, the reachable information range of a node expands rapidly, that is, the size of its $k$-hop subgraph grows exponentially as $k$ increases, leading to a significant increase in the overlap between the $k$-hop subgraphs of different nodes. As a result, the aggregation result of different nodes on their respective $k$-hop subgraphs becomes indistinguishable. Furthermore, in a specific graph dataset, nodes with higher degrees tend to have a larger range of $k$-hop subgraphs compared to nodes with lower degrees. As a result, the subgraphs are more likely to overlap between nodes with higher degrees, making their aggregation results more likely to become similar and indistinguishable.

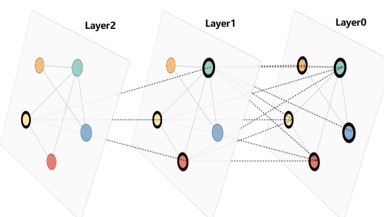

Figure 1: $k$-hop subgraph.

To verify this point, we conduct experiments on three graph datasets, Cora, Citeseer, and Pubmed. First, we group the nodes according to their degrees by assigning nodes with degrees in the range of $[2^i, 2^{i+1})$ to the $i$-th group. Subsequently, we perform aggregation with different layers of GCN and GAT, then calculate the degree of smoothing of the node representations within each group separately. We use the metric proposed in [11] to measure the smoothness of the node representations within each group, namely **SMV**, which calculates the average of the distances between the nodes within the group:

$$\mathbf{SMV}(\mathbf{X}) = \frac{1}{\mathbf{N}(\mathbf{N}-1)} \sum_{i \neq j} \mathbf{D}\left(\mathbf{X}_{i,:}, \mathbf{X}_{j,:}\right) \tag{2}$$

where $\mathbf{D}(\cdot, \cdot)$ denotes the normalized Euclidean distance between two vectors:

$$\mathbf{D}(\mathbf{x}, \mathbf{y}) = \frac{1}{2}\left\| \frac{\mathbf{x}}{\|\mathbf{x}\|} - \frac{\mathbf{y}}{\|\mathbf{y}\|} \right\|_2 \tag{3}$$

A smaller value of **SMV** indicates a greater similarity in node representations.

We select the most representative result illustrated in Figure 2, which shows the result of GAT on Pubmed. The rest of the results are shown in the Appendix B. It can be seen that the groups of nodes with higher degree tend to be more likely to have high similarity in the representation of nodes within the group in different layers of the model. This finding supports our claim.

After verifying the conclusion that subgraph overlap leads to oversmoothing through experiments, a natural idea is to alleviate the problem of large overlap of single subgraph by utilizing multiple hop subgraph aggregations, thereby alleviating oversmoothing. In the following section, we will demonstrate that the previous $k$-layer residual-based GNNs are actually different forms of integration of 1 to $k$ hop subgraph aggregations.

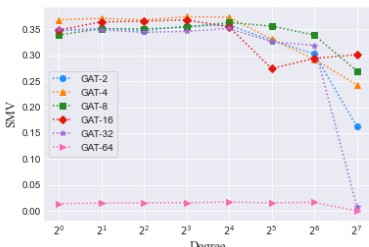

Figure 2: SMV for node groups of different degrees.

### 3.1 Revisiting Previous Models in a New Perspective

In the rest of this paper we will uniformly take GCN, a classical residual-free message-passing GNNs, as an example. We assume that $\mathbf{H}$ is non-negative, so the ELU function can be ignored. In addition, the weight matrix is ignored for simplicity. Combined with the formula of GCN given in Equation 1, we can formulate the specific result of $k$-hop subgraph aggregation as $\mathbf{N}^k\mathbf{H}$, where $\mathbf{N} = \tilde{\mathbf{D}}^{-\frac{1}{2}}\tilde{\mathbf{A}}\tilde{\mathbf{D}}^{-\frac{1}{2}}$. To show more intuitively how different $k$-layer-based residual models utilize $\mathbf{N}^j\mathbf{H}$, $j = 0, 1, \cdots, k$. We derive the general term formulas of their final outputs, and the results are shown in Table 2. Details of the derivation of the formula in this part are given in Appendix C.

Table 2: General term formulas of residual models.

| Model Name | General Term Formula |
|---|---|
| ResGCN | $\mathbf{H}_k = \sum_{j=0}^{k} \mathbf{C}_k^j \mathbf{N}^j \mathbf{H}$ |
| APPNP | $\mathbf{H}_k = (1-\alpha)^k \mathbf{N}^k\mathbf{H} + \alpha \sum_{j=0}^{k-1} \sum_{i=0}^{j} (-1)^{j-i} (1-\alpha)^i \mathbf{N}^i\mathbf{H}$ |
| JKNet | $\mathbf{H}_k = \mathbf{AGG}_{jk}(\mathbf{NH}, \ldots, \mathbf{N}^{k-1}\mathbf{H})$ |
| DenseGCN | — |

From the formula in the table, we can see that, in comparison to message-passing GNNs, residual-based variants of GNNs can utilize multiple $k$-hop subgraphs. There are two methods to exploit them: **(1)** Summation, such as ResGCN and APPNP. Such methods employ linear summation over the aggregation of different hop subgraphs; **(2)** Aggregation functions, such as DenseNet and JKNet. Such methods make direct and explicit exploitation of different hop subgraph aggregations through methods such as concatenation.

However, for the first type of methods, they all employ a fixed, layer-level coefficient for linear summation of the subgraph aggregation, which assumes that the information from the subgraph of the same hop are equally important for different nodes. It will limit the expressive power of GNNs, which reveals the need to design a more fine-grained node-level residual module that can more flexibly utilize information from different $k$-hop subgraphs. For another type of method, they can achieve finer-grained subgraph aggregation, but the experiment find that their performance is not improved because of the more finer-grained structure, mainly because the introduction of more parameters leads to overfitting Phenomenon. In general, neither of these two types of methods has achieved a more effective improvement in the expressive power of GNNs.

## 4 The Proposed Method

In order to solve the two limitations of flexibility and overfitting encountered by previous residual-based models, we try to propose a node-level, more flexible, general residual module, which can

alleviate overfitting caused by more parameters at the same time. Based on this, we propose a sampling-based node-level generic residual module SNR. We define SNR module as:

$$\mathbf{h}_{k-1}^{(i)}{}' = \mathbf{GraphConv}\left(\mathbf{h}_{k-1}^{(i)}\right) \tag{4}$$

$$\mathbf{h}_k^{(i)} = \mathbf{h}_1^{(i)} + \mathbf{sigmoid}(p_{k-1}^{(i)})\left(\mathbf{h}_1^{(i)} - \mathbf{h}_{k-1}^{(i)}{}'\right), \quad p_{k-1}^{(i)} \sim \mathcal{N}(\alpha_{k-1}^{(i)}, \beta_{k-1}^{(i)}{}^2) \tag{5}$$

where $\mathbf{h}_k^{(i)}$ denotes the representation of $k$-th layer of node $i$, $\mathbf{h}_k^{(i)}{}'$ denotes the result obtained by an arbitrary GNNs layer with $\mathbf{h}_{k-1}^{(i)}$ as input. $p_k^{(i)}$ is a random number sampled from $\mathcal{N}(\alpha_k^{(i)}, \beta_k^{(i)}{}^2)$ which associated with the $i$-th node at the $k$-th layer while $\alpha_k^{(i)}$ and $\beta_k^{(i)}$ are learnable parameters representing the mean and the standard deviation of this distribution, respectively. Next, we will illustrate the superiority of the SNR module in terms of flexibility and overfitting alleviation.

## 4.1 Flexibility

In this section, we will analyze the expressive power of GCN with SNR module and show that SNR-GCN achieves a more flexible utilization of multiple subgraph aggregations. First of all, combined with the previous definition 5, the matrix form of the recurrence formula of SNR-GCN can be written as:

$$\mathbf{H}_k = \mathbf{H}_1 + \Lambda_{k-1}\left(\mathbf{H}_1 - \tilde{\mathbf{D}}^{-1/2}\tilde{\mathbf{A}}\tilde{\mathbf{D}}^{-1/2}\mathbf{H}_{k-1}\right) \tag{6}$$

where $\Lambda_k$ is a diagonal matrix whose $i$-th diagonal element is equal to $p_k^{(i)}$. We first try to obtain the general term formula of SNR-GCN according to the recursive formula and demonstrate SNR-GCN's treatment of multiple subgraph aggregations. The following theorem can be proved:

**Theorem 1.** *The general term formula of SNR-GCN can be deduced as:* $\mathbf{H}_k = \sum_{i=2}^{k-1}\prod_{j=i}^{k-1}\tilde{\mathbf{N}}_j\left(\mathbf{M}_i - \mathbf{M}_{i-1}\right) + \prod_{i=1}^{k-1}\tilde{\mathbf{N}}_i\left(\mathbf{H}_1 + \mathbf{M}_1\right) - \mathbf{M}_{k-1}$ *where* $\tilde{\mathbf{N}}_i = -\Lambda_{k-1}\mathbf{N}$ *and* $\mathbf{M}_k = -\left(\Lambda_k\mathbf{N} + \mathbf{I}\right)^{-1}\left(\mathbf{I} + \Lambda_k\right)\mathbf{H}_1$.

The details of the proof are provided in Appendix D. From the general term formula of SNR-GCN, we can see that $\mathbf{M}_k$ is a linear transformation of $\mathbf{H}_1$. Therefore, the first two terms of the formula can be approximately regarded as a new form of subgraph aggregation. Further we can find that all 1 to $k$ hop subgraph aggregations appear in the formula, which ensures the expressive power. And because $\Lambda_k$ are learnable diagonal matrixes , SNR-GCN's subgraph aggregation is learnable and more flexible, which further makes expressive power stronger. Besides, when we set $\Lambda_k = -\alpha\mathbf{I}$, the first term will be 0, and the rest terms are equivalent to APPNP's formula, which means SNR-GCN can be approximately regarded as a more fine-grained and expressive APPNP.

## 4.2 Overfitting Alleviation

Another key point of SNR is that it introduces randomness to alleviate overfitting. In our initial idea, we attempt to build a generic module similar to the initial residual at the node level. Based on this, we initially designed the following modules:

$$\mathbf{h}_k^{(i)} = \mathbf{h}_1^{(i)} + \mathbf{sigmoid}(q_k^{(i)})\left(\mathbf{h}_1^{(i)} - \mathbf{h}_{k-1}^{(i)}{}'\right) \tag{7}$$

where $q_k^{(i)}$ is a learnable parameter which associated with the $i$-th node at the $k$-th layer. After conducting experiments, we discover that the model has a high risk of overfitting when adding this module. However, we also find that if we do not learn $q_k^{(i)}$ directly through backpropagation, but first learn a normal distribution associated with it via reparameterization trick and obtain $q_k^{(i)}$ by sampling at each computation, the issue can be resolved, and the performance of the model significantly improves. To prove this, we perform an experimental verification. The details of the experiments are shown in the Appendix E.

It is worth noting that GCNII and SNR-GCN share a similar architecture, so both can be viewed approximately as more refined APPNP-style models. However, when faced with the problem of overfitting due to more parameters, GCNII adds an identity matrix to mitigate the issue. Later experiment results have shown that SNR-GCN's learning distribution-sampling approach is more effective in alleviating overfitting.

### 4.3 Complexity Analysis

Taking vanilla GCN as an example, we analyzed the additional complexity of SNR in model and time. We assume that the number of nodes in the graph is $n$ and hidden dimension is $d$.

**Model Complexity.** As described in Section 4, at each layer the SNR module learns a mean and standard deviation of the corresponding distribution for each node, so the complexity can be calculated as $O(n)$, and thus the additional complexity of the $k$-layer model equipped with SNR is $O(kn)$.

**Time Complexity.** The time complexity of a vanilla GCN layer mainly comes from the matrix multiplication of $\mathbf{N}$ and $\mathbf{H}$, hence its complexity is $O(n^2 d)$. And the main computational parts of a SNR module are the sampling of $p_k^{(i)}$, scalar multiplication and matrix addition, which correspond to a complexity of $O(n)$, $O(nd)$, and $O(nd)$, respectively. Thus the time complexity of the SNR module is $O(nd)$ and the time complexity of a GCN layer equipped the SNR module is $O(n^2 d + nd)$. Therefore, the introduction of the SNR module does not significantly affect the computational efficiency.

## 5 Experiment

In this section, we aim to experimentally evaluate the effectiveness of SNR on real datasets. To achieve this, we will compare the performance of SNR with other methods and answer the following research questions. **Q1:** How effective is SNR on classical tasks that prefer shallow models? **Q2:** Can SNR help overcome oversmoothing in GNNs and enable the training of deeper models? **Q3:** How effective is SNR on tasks that require deep GNNs? **Q4:** How efficient is the training of SNR?

### 5.1 Experiment Setup

In our study, we conduct experiments on four tasks: semi-supervised node classification **(Q1)**, alleviating performance drop in deeper GNNs **(Q2)**, semi-supervised node classification with missing vectors **(Q3)**, and efficiency evaluation **(Q4)**.

**Datasets.** To assess the effectiveness of our proposed module, we have used four data sets that are widely used in the field of GNN, including Cora, Citeseer, Pubmed [12], and CoraFull [13] for testing purposes. In addition, we also use two webpage datasets collected from Wikipedia: Chameleon and Squirrel [14]. Details on the characteristics of these datasets and the specific data-splitting procedures used can be found in Appendix F.1.

**Models.** We consider two fundamental GNNs, GCN [10] and GAT [15]. For GCN, we test the performance of SNR-GCN and its residual variant models, including ResGCN [9], APPNP [7], DenseGCN [9], GCNII [16] and JKNet [5]. For GAT, we directly equip it with the following residual module: Res, InitialRes, Dense, JK and SNR and test the performance. Additionally, for the SSNC-MV task, we compare our proposed module with several classical oversmoothing mitigation techniques, including BatchNorm [17], PairNorm [18], DGN [19], Decorr [11], DropEdge [20] and other residual-based methods. Further details on these models and techniques can be found in the following sections.

**Implementations.** For all benchmark and variant models, the linear layers in the models are initialized with a standard normal distribution, and the convolutional layers are initialized with Xavier initialization. The Adam optimizer [21] is used for all models. Further details on the specific parameter settings used can be found in Appendix F.2. All models and datasets used in this paper are implemented using the Deep Graph Library (DGL) [22]. All experiments are conducted on a server with 15 vCPU Intel(R) Xeon(R) Platinum 8358P CPU @ 2.60GHz, A40 with 48GB GPU memory, and 56GB main memory.

### 5.2 Semi-supervised Node Classification

To validate the performance of SNR, we apply the module to two fundamental GNNs, GCN and GAT, and test the accuracy according to the mentioned experimental setup, and compare it with four classic residual modules, DenseNet, ResNet, InitialResNet and JKNet. We vary the number of layers in the range of $\{1, 2, 3, \cdots, 10\}$ and select the best result among all layers. Specifically, we run 10 times for each number of layers to obtain the mean accuracy along with the standard deviation. We select the best results among all layers and report them in the Table 3. We find that GNNs with

Table 3: Summary of classification accuracy (%) results with various depths. The best results are in bold and the second best results are underlined.

| Method | Cora | Citeseer | Pubmed | CoraFull | Chameleon | Squirrel |
|---|---|---|---|---|---|---|
| GCN | 80.16±1.15 | 70.20±0.62 | 78.26±0.61 | 68.40±0.33 | 68.00±2.30 | 51.69±1.83 |
| ResGCN | 79.01±1.26 | 69.27±0.66 | 78.08±0.51 | 67.98±0.51 | 65.26±2.47 | 47.43±1.14 |
| APPNP | 79.04±0.84 | 69.64±0.49 | 76.38±0.12 | 37.77±0.43 | 59.80±2.68 | 43.17±1.01 |
| GCNII | 78.53±0.67 | 69.55±1.14 | 76.17±0.70 | 68.30±0.26 | 64.76±2.43 | 52.83±1.51 |
| DenseGCN | 77.24±1.12 | 65.03±1.58 | 76.93±0.78 | 64.52±0.71 | 59.04±2.07 | 38.89±1.25 |
| JKNet | 78.16±1.21 | 65.33±1.66 | 78.10±0.55 | 66.11±0.49 | 55.75±2.93 | 35.95±1.10 |
| **SNR-GCN (Ours)** | **81.17±0.72** | **70.39±1.01** | **78.34±0.62** | **69.80±0.28** | **72.04±1.89** | **58.35±1.55** |
| GAT | 79.24±1.18 | 69.51±1.07 | 77.59±0.80 | 67.39±0.32 | 65.81±2.13 | 50.16±2.42 |
| Res-GAT | 78.43±0.99 | 68.15±1.25 | 77.27±0.52 | 67.67±0.32 | 69.08±2.50 | 49.77±1.72 |
| InitialRes-GAT | 77.77±1.51 | 67.48±2.15 | 77.46±1.17 | 65.49±0.42 | 65.90±2.98 | 52.83±2.39 |
| Dense-GAT | 78.27±2.22 | 64.92±1.94 | 76.84±0.64 | 66.61±0.63 | 63.86±3.03 | 43.01±1.34 |
| JK-GAT | 78.91±1.71 | 65.59±2.62 | 77.70±0.64 | 67.69±0.65 | 56.14±2.68 | 37.25±1.01 |
| **SNR-GAT (Ours)** | **79.65±0.84** | **69.85±0.67** | **77.76±0.93** | **68.00±0.27** | **69.54±2.22** | **55.14±1.78** |

the SNR module consistently achieve the best performance in all cases **(Q1)**. However, from the experimental results, many models with residual modules have not achieved the expected results. In many cases, compared with the basic model, the accuracy is even reduced. According to previous research [18], we speculate that overfitting may have contributed to this phenomenon. To verify our hypothesis, we conduct further experiments. Given that most models in the previous experiments achieve their best performance with shallow models, we select models with two layers, train 500 epochs, and report their accuracy on the training and validation sets at each epoch. The results are shown in Appendix G. Most models show signs of overfitting and SNR module demonstrates the best ability to alleviate overfitting. Specifically, in shallow GNNs with limited subgraph aggregation, most models have similar expressive abilities, and overfitting is the main factor affecting their performance. Our proposed method effectively alleviates overfitting by learning a more representative distribution, resulting in a better performance than the base models.

## 5.3 Alleviating Performance Drop in Deeper GNNs

As the number of layers in GNNs increases, oversmoothing occurs, resulting in performance degradation. Our objective is to investigate the performance of deep GNNs equipped with SNR and observe the impact of oversmoothing on their performance. We evaluate the performance of GNNs with different residual modules on 2, 16, and 32 layers using the Cora, Citeseer, and Pubmed datasets. The "None" column represents vanilla GNNs without any additional modules. According to [16], APPNP is a shallow model, hence we use GCNII to represent GCN with initial residual connection instead. The same settings are used in section 5.4. The experimental results are presented in Table 4.

From Table 4, we can observe that GNNs with SNR consistently outperform other residual methods and the base models in most of cases when given the same number of layers. SNR can significantly improve the performance of deep GNNs **(Q2)**. For instance, on the Cora dataset, SNR improves the performance of 32-layer GCN and GAT by **53.69%** and **56.20%**, respectively. By flexibly utilizing multiple subgraph aggregation results with our SNR module, we can enhance the expressive power of the model and produce more distinctive node representations than those of regular GNNs, thereby overcoming the oversmoothing problem. These results suggest that we can train deep GNNs based on SNR, making them suitable for tasks that require the use of deep GNNs.

## 5.4 Semi-supervised Node Classification with Missing Vectors

When do we need deep GNNs? [18] first proposed semi-supervised node classification with missing vectors (SSNC-MV), where nodes' features are missing. SSNC-MV is a practical problem with various real-world applications. For example, new users on social networks usually lack personal information [23]. Obviously, we need more propagation steps to effectively aggregate information associated with existing users so that we can obtain representations of these new users. In this scenario, GNNs with more layers clearly perform better.

Table 4: Node classification accuracy (%) on different number of layers. The best results are in bold and the second best results are underlined.

| Dataset | Method | GCN | | | GAT | | |
|---|---|---|---|---|---|---|---|
| | | L2 | L16 | L32 | L2 | L16 | L32 |
| Cora | None | 79.50±0.84 | 69.83±2.47 | 25.31±12.49 | 79.11±1.55 | 75.44±1.08 | 22.74±7.47 |
| | Res | 78.73±1.27 | 78.46±0.79 | 38.70±8.20 | 78.36±1.42 | 34.80±6.26 | 32.06±0.54 |
| | InitialRes | 77.67±0.51 | 77.74±0.73 | 77.92±0.56 | 77.20±1.54 | 74.99±0.75 | 25.08±7.27 |
| | Dense | 75.24±1.73 | 71.34±1.51 | 75.43±2.49 | 76.80±1.71 | 74.75±2.22 | 75.70±2.20 |
| | JK | 76.28±1.73 | 72.39±3.20 | 75.03±1.11 | 78.06±0.51 | 76.66±1.39 | 23.29±8,45 |
| | SNR (Ours) | 80.58±0.82 | 78.55±0.92 | 79.00±1.43 | 79.69±0.55 | 77.92±1.54 | 78.94±0.80 |
| Citeseer | None | 68.31±1.40 | 54.07±2.48 | 34.84±1.60 | 68.64±1.20 | 59.16±2.44 | 24.37±3.59 |
| | Res | 67.68±1.36 | 63.99±1.12 | 25.96±4.27 | 67.55±1.10 | 28.53±4.93 | 24.70±4.12 |
| | InitialRes | 68.23±0.95 | 68.29±0.92 | 68.74±0.61 | 66.86±1.60 | 60.24±2.29 | 23.78±4.87 |
| | Dense | 64.83±0.94 | 58.42±2.96 | 58.75±3.37 | 64.58±2.07 | 61.17±1.78 | 61.87±2.91 |
| | JK | 64.69±1.44 | 58.38±3.36 | 58.63±4.76 | 65.84±2.02 | 62.64±1.66 | 23.09±4.02 |
| | SNR (Ours) | 70.18±0.61 | 67.07±1.78 | 66.27±2.00 | 69.71±0.92 | 67.51±2.28 | 66.53±2.48 |
| Pubmed | None | 77.53±0.73 | 76.16±0.96 | 51.29±11.71 | 77.07±0.52 | 77.49±0.65 | 53.20±9.18 |
| | Res | 77.64±1.01 | 77.65±0.78 | 73.31±7.15 | 77.36±0.60 | 50.16±7.65 | 43.46±3.30 |
| | InitialRes | 75.66±0.82 | 75.15±0.48 | 75.31±0.55 | 77.42±0.79 | 77.42±0.82 | 44.96±5.91 |
| | Dense | 76.81±1.06 | 74.01±2.36 | 76.33±1.17 | 76.66±0.61 | 76.38±1.26 | 76.50±1.47 |
| | JK | 77.61±0.78 | 76.31±1.45 | 76.59±1.53 | 77.48±0.84 | 77.75±0.77 | 40.84±0.23 |
| | SNR (Ours) | 77.84±0.51 | 78.02±0.71 | 77.36±0.78 | 77.51±0.62 | 78.17±0.85 | 77.77+0.46 |

Table 5: Test accuracy (%) on missing feature setting. The best results are in bold and the second best results are underlined.

| | GCN | | | | | | GAT | | | | | |
|---|---|---|---|---|---|---|---|---|---|---|---|---|
| Method | Cora | | Citeseer | | Pubmed | | Cora | | Citeseer | | Pubmed | |
| | Acc | #K | Acc | #K | Acc | #K | Acc | #K | Acc | #K | Acc | #K |
| None | 57.3 | 3 | 44.0 | 6 | 36.4 | 4 | 50.1 | 2 | 40.8 | 4 | 38.5 | 4 |
| BatchNorm | 71.8 | 20 | 45.1 | 25 | 70.4 | 30 | 72.7 | 5 | 48.7 | 5 | 60.7 | 4 |
| PairNorm | 65.6 | 20 | 43.6 | 25 | 63.1 | 30 | 68.8 | 8 | 50.3 | 6 | 63.2 | 20 |
| DGN | 76.3 | 20 | 50.2 | 30 | 72.0 | 30 | 75.8 | 8 | 54.5 | 5 | 72.3 | 20 |
| DeCorr | 73.8 | 20 | 49.1 | 30 | 73.3 | 15 | 72.8 | 15 | 46.5 | 6 | 72.4 | 15 |
| DropEdge | 67.0 | 6 | 44.2 | 8 | 69.3 | 6 | 67.2 | 6 | 48.2 | 6 | 67.2 | 6 |
| Res | 74.06±1.10 | 7 | 57.52±1.30 | 6 | 76.32±0.41 | 8 | 74.86±1.25 | 6 | 57.88±2.79 | 4 | 76.70±0.55 | 7 |
| InitialRes | 60.68±1.29 | 2 | 46.86±4.14 | 10 | 69.14±0.90 | 7 | 60.68±1.29 | 2 | 57.34±3.78 | 4 | 76.10±0.70 | 4 |
| Dense | 70.52±3.21 | 10 | 54.96±2.25 | 9 | 75.26±1.32 | 8 | 70.52±3.21 | 10 | 58.28±0.14 | 10 | 75.22±1.21 | 15 |
| JK | 72.68±2.61 | 8 | 57.54±1.14 | 10 | 76.44±1.51 | 20 | 72.68±2.63 | 8 | 58.82±2.02 | 5 | 76.12±0.87 | 10 |
| SNR (Ours) | 76.34±0.68 | 7 | 61.78±1.41 | 9 | 76.92±0.70 | 8 | 77.02±0.89 | 9 | 61.00±1.07 | 8 | 77.00±0.74 | 20 |

Previous research has shown that normalization techniques can be effective in mitigating oversmoothing, and further, exploring deeper architectures. Therefore, we apply several techniques that can overcome oversmoothing and residual modules to GCN and GAT to compare their performance on tasks that require deep GNNs.

We remove the node features in the validation and test set following the idea in [11, 18, 19]. We reuse the metrics that already reported in [11] for None, BatchNorm [17], PairNorm [18], DGN [19], DeCorr [11], and DropEdge [20]. For all residual-based models, the results are obtained by varying the number of layers in $\{1, 2, 3, \cdots, 10, 15, \cdots, 30\}$ and running five times for each number of layers. We select the layer #K that achieves the best performance and report its average accuracy along with the standard deviation. The results are reported in Table 5.

Our experiments show that GNNs with the SNR module outperform all previous methods (**Q3**). Additionally, we find that for most models, the number of layers to reach the best accuracy is relatively large, which indicates that it is necessary to perform more propagation to gather information from further nodes so that we can obtain effective representations of nodes with missing features.

## 5.5 Efficiency Experiment

In real-world tasks, the rate at which a model achieves optimal performance through training is often important, and this affects the true effectiveness and time consumption of the model in real-world applications. To enable concrete measurement and comparison, here we define the following metrics for model training efficiency:

$$\textbf{Efficiency} = \frac{\textbf{Accuracy}}{\textbf{Time}} \tag{8}$$

where **Accuracy** denotes the accuracy of the model when it reaches its optimal performance and **Time** denotes the time when the model reaches its optimal performance. The definition of this formula shows that a larger **Efficiency** represents a higher performance per unit time improvement, and therefore a higher training efficiency.

Based on the above equation, we evaluate the training efficiency of vanilla GNNs and SNR-GNNs. We use the 2, 4, 8, 16, 32, and 64-layer and average five **Efficiency** calculated for each layer of the model. Specifically, each **Efficiency** is calculated based on the time for the model to reach the highest accuracy on the validation set after 100 epochs of training and the accuracy achieved on the test set at that time. Figure 3 shows the models' **Efficiency** on Cora. The results on other datasets are shown in the Appendix H. It can be noticed that the training efficiency decreases as the number

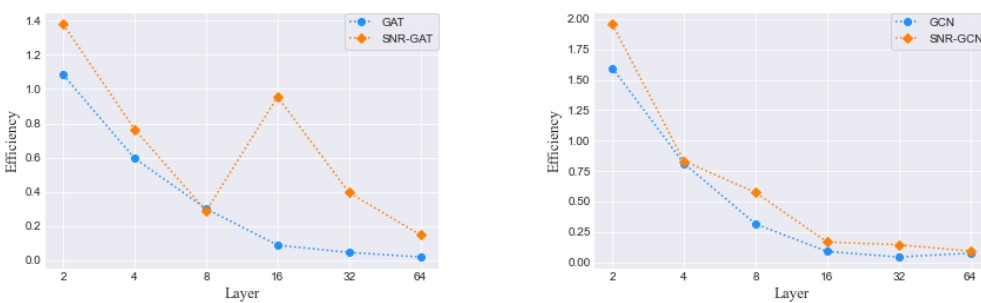

Figure 3: Efficiency for different models at different layers.

of layers increases, which is due to the increase in training time caused by the rise in the number of model parameters. However, in most cases, compared to vanilla GNNs, our SNR module is able to maintain the highest training efficiency **(Q4)**.

## 6 Conclusion

Our work proposes a new perspective for understanding the expressive power of GNNs: the *k*-hop subgraph aggregation theory. From this perspective, we have reinterpreted and experimentally validated the reason why the performance of message-passing GNNs decreases as the number of layers increases. Furthermore, we have evaluated the expressive power of previous residual-based GNNs based on this perspective. Building on these insights, we propose a new sampling-based generalized residual module SNR and show theoretically that SNR enables GNNs to more flexibly utilize information from multiple *k*-hop subgraphs, thus further improving the expressive power of GNNs. Extensive experiments demonstrate that the proposed SNR can effectively address the issues of overfitting in shallow layers and oversmoothing in deep layers that are commonly encountered in message-passing GNNs, and significantly improves the performance, particularly in SSNC-MV tasks. Our research will facilitate a deeper exploration of deep GNNs and enable a wider range of potential applications.

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
