# OpenReview forum: "Deep Graph Neural Networks via Flexible Subgraph Aggregation"
_NeurIPS.cc/2023/Conference — Submitted to NeurIPS 2023_

### Official Review · Reviewer_EwhV · 2023-07-06

**Soundness:** 3 good
**Presentation:** 3 good
**Contribution:** 4 excellent
**Rating:** 7
**Confidence:** 4

**Summary:**

This manuscript understands the expressive power of GNNs from a new perspective of subgraph aggregation, and reveals the potential reason for the performance degradation of traditional deep GNNs due to the overlap of aggregated subgraphs. The authors propose a sampling-based generalized residual module SNR and theoretically proves that SNR enables GNNs to more flexibly utilize information from multiple k-hop subgraphs, thereby improving the expressive power of GNNs. Extensive experiments show the effectiveness of the proposed SNR module.

**Strengths:**

1）The idea of rethinking the expressive power of GNNs from the perspective of subgraph aggregation is novel and interesting.

2）The paper is well presented. The motivation, steps to construct a node-level, more flexible and general residual module to enhance the expressive power of GNNs while alleviating overfitting issue are clearly introduced.

3）The experimental results are convincible.


**Weaknesses:**

1）More related works on dealing with oversmoothing and overfitting issues in deep graph neural networks should be reviewed.

2）Further analysis of the experimental results is needed. For example, in Table 4, the possible reason why the proposed SNR module is weaker than InitialRes on Citeseer dataset should be discussed.

3）There are some typos, e.g., row 221: “four data sets” should be “six datasets”.


**Questions:**

1）Why did the author use different backbone models on Table 3 and Table 4? GCNII is not used in Table 4.

2）What is the full name of the evaluation metric “SMV”.


**Limitations:**

Yes.

---

> ### Author Rebuttal · Authors · 2023-08-07
>
> Thank you very much for your valuable feedback. Below are our responses to your questions/concerns.
>
> > More related works on dealing with oversmoothing and overfitting issues in deep graph neural networks should be reviewed.
> >
>
> Thanks for the useful comment. We will discuss more related work on oversmoothing and overfitting issues in our related work and preliminaries. For example, we will add [1] which related oversmoothing to the stability of steady states of the underlying ODE and proposed a framework called Graph-Coupled Oscillator Network (GraphCON), [2] which analyzed the bottleneck of deep GNNs by leveraging the Dirichlet energy of node embeddings and designed Energetic Graph Neural Networks (EGNN) based on their theoretical results, [3] which relieved the oversmoothing issue by optimizing the graph topology to make it more suitable for downstream tasks and proposed two methods from the topological view.
>
> > Further analysis of the experimental results is needed. For example, in Table 4, the possible reason why the proposed SNR module is weaker than InitialRes on Citeseer dataset should be discussed.
> >
>
> We observed that on the Citeseer dataset, GCN with Initial Residual outperforms our module with 16 and 32 layers. First of all, it can be noticed that no matter in which dataset, GCN with initial Residual, that is, GCNii, maintains an unsatisfactory but relatively stable result, while GAT with Initial Residual (no identity matrix) does not have this stable result. As a result, we speculate that the identity matrix of GCNii maintains the stability of the multi-layer results at the expense of the overall performance to a certain extent. Our module shows a large performance degradation with 16 layers on Citeseer, and the performance with 32 layers is close to that of 16 layers, so we believe that it may be because on the Citeseer dataset, our module has overfitted in multiple layers, resulting in performance degradation and lower than the results of GCNii. Although Performance degradation has occurred, our method still far outperforms all other methods except for GCNii, so we feel that this phenomenon does not affect the demonstration of SNR performance.
>
> > There are some typos, e.g., row 221: “four data sets” should be “six datasets”.
> >
>
> Thanks for bringing this to our attention. In fact, when referring to “four datasets”, we meant the fundamental datasets Cora, Citeseer, Pubmed, and CoraFull. Additionally, we provided information about the two supplementary wiki datasets Chameleon and Squirrel in Line 223. We will thoroughly review the paper again and rectify these confusions.
>
> > Why did the author use different backbone models on Table 3 and Table 4? GCNII is not used in Table 4.
> >
>
> Actually, we did use the GCNII model in Table 4, but we confusingly referred to it as ‘InitialRes’ in the table. Additionally, we would like to note that we have mentioned this in line 266 but the original wording may not have been clear enough. We apologize for this confusion and we will make sure to change the name of InitialRes to GCNII in Table 4 and clarify this in the revision.
>
> > What is the full name of the evaluation metric “SMV”.
> >
>
> The full name of SMV is smoothness metric value, which is a smoothness metric proposed in [4]. We will clarify this in the revision.
>
> We hope that our explanations above have addressed all of the reviewer's concerns, and we are happy to answer any further questions.
>
> **References**
> [1]Rusch, T. Konstantin, et al. “Graph-Coupled Oscillator Networks.” *International Conference on Machine Learning*, 2022.
> [2]Zhou, Kaixiong, et al. “Dirichlet Energy Constrained Learning for Deep Graph Neural Networks.” Neural Information Processing Systems, 2021.
> [3]Chen, Deli, et al. “Measuring and Relieving the Over-Smoothing Problem for Graph Neural Networks from the Topological View.” Proceedings of the AAAI Conference on Artificial Intelligence, vol. 34, no. 04, 2020, pp. 3438–45.
> [4]Liu, Meng, et al. “Towards Deeper Graph Neural Networks.” Proceedings of the 26th ACM SIGKDD International Conference on Knowledge Discovery & Data Mining, ACM, 2020.

---

> ### Comment · Reviewer_EwhV · 2023-08-18
> **Response to the rebuttal**
>
> The author's rebuttal has addressed my main comments. I like the idea of this work that uses subgraph aggregation to alleviate the oversmoothing and overfitting problem of GNNs. It may give some inspiration from the perspective of the expressive power of GNNs. But I also wonder about the limitation of this approach, i.e., when it works and when it does not work. It is suggested that the authors discuss the applicability and limitation of this work.

---

> > ### Author Response · Authors · 2023-08-19
> > **Thanks for the reply and clarifications on applicability/limitation**
> >
> > We thank the reviewer for acknowledging the interestingness of our idea and the potential inspiration from our work. In the following, we provide further clarifications regarding the applicability and limitation of this work.
> >
> > The major applicable scenarios of our method are semi-supervised node classifications where deeper GNNs are needed, for example, when node attributes are noisy/incomplete (for example, the SSNC-MV scenario mentioned in the paper and multiple existing studies on deep GNNs) and/or when subtle graph structures matter (for example, on graphs where node labels are highly related with the structural roles such as Chamelon and Squirrel). The **clear advantages** of our method in these scenarios can be particularly verified by our experimental results in Table 3 and Table 5 in the main paper. On the contrary, graphs that have rather complete/accurate node features and/or rather simple structures would typically not need deeper GNNs, and thus the advantages of our method would be **relatively limited** for them.
> >
> > Since the focus of this work is semi-supervised node classification, we have not systematically tested our method or some simple extensions of it on other tasks such as inductive learning (as also pointed out by reviewer 5Mqj), link prediction, and graph classification. However, we would rather not call these scenarios limitations of our method. Instead, they are currently unknown and provide valuable directions for future work.
> >
> > We thank the reviewer again for raising this suggestion and we are happy to include discussions as such in our experiments and conclusions sections in the revision.

---

> ### Comment · Reviewer_EwhV · 2023-08-19
>
> I appreciate the comprehensive responses as well as the potential inspiration of this work. Thus, I still support the acceptance of this paper and can also raise my score if needed.

---

> > ### Author Response · Authors · 2023-08-20
> >
> > We thank the reviewer for the confirmed support. An increased score would be certainly appreciated!

---

### Official Review · Reviewer_f2R8 · 2023-07-06

**Soundness:** 2 fair
**Presentation:** 3 good
**Contribution:** 2 fair
**Rating:** 5
**Confidence:** 4

**Summary:**

The paper propose a sampling-based module to enhance the expressive power of Graph Neural Networks (GNNs), which traditionally assume that information from the subgraph of the same hop is equally important for all nodes in the graph. They argue that this rigid assumption restricts the models' ability to capture complex relationships. With their proposed module, different nodes can assign varying levels of importance to their neighbors at different hops during information aggregation, which is shown to improve the performance of the GNN variants.

**Strengths:**

1.	The proposed idea is interesting.
2.	The paper is well written and clear.
3.	The method is described in sufficient detail and easy to follow.


**Weaknesses:**

1.	Some reported baseline methods’ performances are questionable. For example, GCNII [1] shows alleviation of over-smoothing on the benchmark datasets with even 32 and 64 layers and achieved much higher accuracy than reported in Table 3.
2.	The experiments are mainly performed on small-scale graph datasets. It’ll be interesting to test the method on larger graph datasets (e.g., the OGB datasets).
3.	Some detailed analysis of the learned mean and variance of the normal distribution would better support the proposed idea. It would be interesting to see how the distribution for nodes of different degrees changes with the increasing number of layers.
4.	To evaluate the method’s robustness to oversmoothing, it will be better to see a plot about how the performance changes as the number of layers increases than presenting the performance in a table. What’s more, the proposed method’s performance significantly dropped as the layer number increased. It suggests it still suffers from oversmoothing and/or overfitting.
5.	It’ll be interesting to see the over-smoothing analysis in Figure 2 to be done for different GCN variants with and without the proposed module.

References:
[1] Chen, Ming, et al. "Simple and deep graph convolutional networks." International conference on machine learning. PMLR, 2020.


**Questions:**

1.	Was the mean and variance parameters of the normal distribution regularized by using any prior?
2.	Why is GCNII not included in comparison in table 4? Although it is claimed in line 266 that APPNP is a shallow model, GCNII is used to represent GCN with initial residual connection instead. But GCNII is not only about initial residual connection and has significant performance improvements over other models, so the comparison seems not proper.
3.	What is the rationale to sample p_k from a normal distribution and pass it to a sigmoid function in equation 5? Would directly sampling p_k from a beta distribution do the work? Since its outputs are in the range 0 to 1 and also has a reparameterization form based on a digamma function.


**Limitations:**

yes

---

> ### Author Rebuttal · Authors · 2023-08-07
>
> We thank the reviewer very much for the valuable feedback. Below are our responses to your questions/concerns.
>
> >**W 1:**
>
> Thank you for the comment. We have used the official DGL implementation of GCNII with the same parameter settings as in the original paper, and we have limited the number of layers to 1-10 in our experiments, as reported in Table 3 in the original draft. We have also conducted experiments with deeper layers, with the results reported in Table 4 in the original draft.
>
> >**W 2:**
>
> We agree that testing our method on larger graph datasets would be interesting and informative. Due to time constraints, we could only conduct some initial experiments during the rebuttal, e.g. on OGB-arxiv. We use the hyperparameters for GCN provided by the official OGB code, run the experiments for 20 times, and present the results in Table B in the new pdf file. The results did show some improvements brought by our method. We will include complete experimental results in the revision.
>
> >**W 3:**
>
> This is a great idea. We have plotted figures of the variance and mean of the distribution with respect to the number of layers and they are shown in Figure A in the new pdf file. We divided all the nodes into four groups according to the degree range, i.e., Group 1 to 4 in the figure, and then recorded the mean values of Mean and Std corresponding to the nodes of each group at different layers.
> It can be found that when the number of layers is small, the learned mean and variance have always been a stable value. As the number of layers increases to a certain extent, a sudden peak will appear. We will add discussions about this phenomenon in the appendix and further explore its implications in our future work.
>
> >**W 4:**
>
> This is also a great suggestion. We will present more results using plot figures instead of tables, especially for presenting the results along with changes of GNN layers. In addition, our module can alleviate the oversmoothing problem significantly better than existing residual-based methods, but not completely solve the oversmoothing problem.
>
> >**W 5:**
>
> This is a good idea. Based on your suggestion, we conducted the same experiment as Figure 2 in the original draft using SNR-GAT, and the results are shown in Figure D in the new pdf file. Compared with Figure 2, it can be seen that GAT with the SNR module can effectively alleviate over-smoothing, especially when the number of model layers is large, for example, with 64 laters. In addition, we also conduct experiments with 64 layers of GAT and SNR_GAT on the Pubmed dataset, and the experimental results are 41.24±0.28 and 77.58±0.95, respectively. Combined with Table 4 in the original draft, we can find that the performance of GAT suffers from a serious degradation starting from around 32 layers, while the performance of our method maintains a very high level after 32 layers, and at the same time, compared with the previous layers, the performance improvement is more significant. This can corroborate results in Figure 2, which verifies that our method can improve the model performance by mitigating oversmoothing.
>
> >**Q 1:**
>
> In our method, we employ the reparameterization technique to learn variance and mean parameters through backpropagation without explicitly utilizing any prior information for regularization.
>
> >**Q 2:**
>
> Apologies for the unclear statement in L266. What we actually mean is this particular baseline named InitialRes in Table 4 is exactly the GCNII method. We will make sure to clarify this in the revision and change the name of GCNII in Table 4 to avoid similar confusion.
>
> >**Q 3:**
>
> We thank the reviewer for putting forward this new point of view. We chose this sampling method to extract a random number in the range of 0~1 as the residual coefficient of the node. Under our research, we found that the probability density curve obtained by this sampling method is very similar to the beta distribution. We drew some classic beta distribution curves and the distribution curve obtained by sampling in our paper in Figure B in the new pdf file. Based on this comparison, it should be possible to directly sample from a beta distribution in our method.
>
> In our method, we chose to sample from a normal distribution because we can directly utilize the standard reparameterization trick for the optimization, so the parameters can be directly decoupled from the distribution, which can be easily combined into the backpropagation algorithm of the neural network to update the parameters according to the gradient. Referring to a paper by Ruiz, Francisco J. R., et al. [1], the beta distribution does not admit the standard reparameterization trick, but it has a generalized form of reparameterization based on the digamma function:
>
> $$
> \epsilon=\mathcal{T}^{-1}(z ; \alpha, \beta)=\frac{\operatorname{logit}(z)-\psi(\alpha)+\psi(\beta)}{\sigma(\alpha, \beta)}.
> $$
>
> In contrast, the parameters cannot be completely decoupled from the sampling, and Monte Carlo simulation is required to estimate the gradient, so the sampling method in this paper is more concise.
>
> We hope that our explanations above have addressed all concerns of the reviewer. We are happy to answer any further questions.
>
> **Reference**
> [1] Ruiz, Francisco J. R., et al. “The Generalized Reparameterization Gradient.” NIPS, 2016.

---

> > ### Author Response · Authors · 2023-08-17
> > **Author's follow-up**
> >
> > Dear reviewer f2R8,
> >
> > We understand that chasing down your reply is not our job and we do not intend to add any pressure on your busy schedule. However, as we are getting closer to the end of the discussion phase, we would really appreciate it if you could be so kind to let us know if we have properly addressed your comments and questions in the rebuttal, and if anything can be further clarified.
> >
> > Many thanks in advance!
> >
> > Authors

---

> ### Comment · Reviewer_f2R8 · 2023-08-18
>
> I appreciate the authors' reply and additional results, after reading other reviews, I will keep my current rating.

---

> > ### Author Response · Authors · 2023-08-18
> > **Thanks for the reply**
> >
> > We thank the reviewer for the reply and we will properly incorporate the discussions in the rebuttal into our revised paper.

---

### Official Review · Reviewer_M5qJ · 2023-07-07

**Soundness:** 2 fair
**Presentation:** 2 fair
**Contribution:** 2 fair
**Rating:** 4
**Confidence:** 4

**Summary:**

This paper studies the problem of over-smooth with GNNs. It is argued that the over-smooth problem is caused by the increased overlap of the sub-graph when the respective field of GNNs becomes larger and larger.  In order to alleviate this problem, this paper proposed a method that random weighting the nodes within each layer by node-wise and layer-wise learnable parameters. The proposed method is limited evaluated on several datasets.

**Strengths:**

- Introducing randomness by using re-parameterization trick is interesting and makes sense to alleviate overfitting.

- Formulating different types of residual GNNs from a unified view is good.

**Weaknesses:**

- Some statements from this manuscript do not stand well. For example, it is argued from the 98-th line that as the k increases, the overlap between k-hop subgraph rises, making the aggregation from the k-hop subgraph from different nodes indistinguishable. It is not evident and the overlap is not sufficient for the over-smooth issues. Think about the transformer architecture where the self-attention within each layer has access to all other nodes/tokens while performing well on pixel-level tasks with several layers.

- Additionally, the experiment that nodes with higher degrees tend to have more similar representations seem can not support the claim of sub-graph theory, as the overlap of the sub-graph does not influence by its degree.

- As for the over-smoothing problem, there are many other methods like drop-edge, which also introduce randomness when training the GNNs. It is highly recommended to include comparisons in the main results with methods along this line to make the contribution clearer (rather than just some comparisons under the setup of missing vectors).

- It is unclear how the proposed method can help under the setup of missing vectors rather than other methods.

- As the sampling parameters are node-wise, I'm wondering how the proposed method can extend to inductive learning.

Some minor issues:

- Missing punctuation at the end of all equations.

- The quality of the figures can be improved a lot. It is highly recommended to use vector graphics for all the visualization.

**Questions:**

Please see the weakness section.

**Limitations:**

There is no limitation discussion. The authors are encouraged to include as least the discussions about the limitation for inductive learning.

---

> ### Author Rebuttal · Authors · 2023-08-07
>
> Thank you very much for your valuable feedback. Below are our responses to your questions/concerns.
>
> >**W1:**
>
> Firstly, our work focuses on the over-smoothing problem of classical MPNNs, which are localized graph models based on the massage-passing mechanism, and are much more scalable than global transformer models. However, MPNNs do not have a positional encoding that encodes the graph structure like transformers to directly utilize the topological information of the graph, nor do they have an effective self-attention mechanism for graph-structured data that guarantees the expressiveness and distinguishability of the aggregated results on a larger or even global information domain after many iterations, and thus need to use residuals and other methods to ameliorate this drawback. Based on this, we believe that it is inappropriate to refute our subgraph overlap theory by using transformers as an example. We have also empirically verified the effectiveness of our subgraph theory.
>
> Secondly, we believe the reviewer might be confusing the concept of vision tasks with graph tasks. Tokens or pixel-level tasks are not of the same nature as the semi-supervised node classification task studied in this work.
>
> >**W2:**
>
> Thank you for pointing this out but we don’t agree with this viewpoint. We believe that nodes with higher degrees tend to aggregate more neighbors in each GNN layer than nodes with lower degrees. Consequently, nodes with higher degrees are more likely to have larger subgraph boundaries, and this difference between subgraph boundaries will further increase along with deeper GNN layers. Therefore, it can be argued that nodes with larger degrees, under the same number of aggregation steps, are more inclined to have larger subgraphs compared to nodes with smaller degrees.
>
> Moreover, in the graph, larger subgraphs tend to have a higher degree of overlap compared to smaller subgraphs. This can be illustrated through an extreme scenario. Let’s consider a situation where several larger subgraphs initially expand to cover the entire graph. In this case, these larger subgraphs will have complete overlap, while smaller subgraphs may be dispersed throughout different parts of the entire graph, resulting in a relatively lower degree of overlap and making it less likely for them to have complete overlap.
>
> We also performed experiments to validate this. The nodes are divided into 7 groups according to the degree range, and the subgraph overlap of nodes with different degree ranges is tested. The overlap degree D is calculated as D = N/(N1+N2-N), where N denotes the number of overlapping nodes, and N1, N2 denote the number of nodes in the two subgraphs. The results are plotted in Figure C in the new pdf file. It can be seen that as the number of model layers increases or the node degree increases, the overlap situation has a clear increasing trend, which verifies our argument.
>
> In conclusion, as the overlap can be reflected through the degree, the experimental results illustrated in Figure 2 in our original draft can provide support for our subgraph theory.
>
> >**W3:**
>
> In our work, our module addresses the over-smoothing problem from a residual perspective, and as a result, the paper places more emphasis on comparing and analyzing it against other residual-oriented methods. Moreover, dropedge introduces randomness essentially for data augmentation purposes, while in our method, some parameters of the model are inherently stochastic. Hence, these two approaches of introducing randomness are not equivalent. In response to the reviewer's concern, we have extended experiments on dropedge. We use GCN as the base model to see the improvement effect of dropedge and SNR on the model, where we use the official code of dropedge from the DGL library, and set the ratio of dropedge to 0.1, 0.3, 0.5, 0.7. The results of 20 runs are averaged. The results are shown in Table A in the new pdf file. As can be seen from the table, our residual module is generally more effective than dropedge, without the need to select the key hyperparameter of dropout ratio.
>
> >**W4:**
>
> Semi-supervised node classification with missing vectors (SSNC-MV) was first proposed in [1] as a scenario where deeper graph neural networks have more advantages. In this scenario, the missing features of some nodes lead to the need for a deeper GNN to aggregate more neighborhood information. The two facts that our method can effectively enable deeper GNNs and it has strong performance in the SSNC-MV setting are thus mutually supportive. Similar observations and conclusions were also implied in other existing works on deep GNNs [2,3].
>
> >**W5:**
>
> The focus of our paper is transductive semi-supervised node classification, but it is an interesting idea to extend our method to the inductive setting, which can be achieved by employing some simple techniques. For instance, we can average the learned residual coefficients for nodes with similar degrees in the training data and assign these coefficients to nodes in the test graph based on their degrees. We intend to further investigate this idea in our future work.
>
> >**W6:**
>
> Thanks for the correction. We will revise the manuscript to include the necessary punctuation marks at the end of each equation in the final version.
>
> >**W7:**
>
> Thanks for the suggestion. We will use vector graphics for all figures in the revision.
>
> We hope that our explanations above have addressed all concerns of the reviewer. We are happy to answer any further questions.
>
> **References**
> [1] Zhao, Lingxiao, et al. “PairNorm: Tackling Oversmoothing in GNNs.” International Conference on Learning Representations, 2019.
> [2] Jin, Wei, et al. “Feature Overcorrelation in Deep Graph Neural Networks.” Proceedings of the 28th ACM SIGKDD Conference on Knowledge Discovery and Data Mining, ACM, 2022.
> [3] Zhou, Kaixiong, et al. “Towards Deeper Graph Neural Networks with Differentiable Group Normalization.” NIPS, 2020.

---

> ### Comment · Area_Chair_ixaq · 2023-08-13
> **Please respond to the rebuttal**
>
> Dear reviewer,
>
> You have given the paper the lowest score. Please read the rebuttal and indicate whether it has addressed your concerns.
>
> Thanks!

---

> ### Comment · Reviewer_M5qJ · 2023-08-14
> **Thanks authors for the responses**
>
> I appreciate the authors for providing detailed responses to address the concerns I raised. While some of the issues have been successfully resolved, there remain several outstanding concerns.
>
> - One significant limitation of the proposed method is its reliance on the transductive setup. This constraint considerably restricts the potential applications of the method. For instance, in tasks like protein property prediction. Although the authors suggest a potential solution, its validity cannot be unverified without comprehensive experimental validation.
>
> - Reviewer f2R8 pointed out that the experiments have primarily been conducted on small datasets. While the authors have conducted an experiment on the OGB-Arxiv dataset using GCN as the base model, the choice of a relatively weak baseline model like GCN raises concerns about evaluating the true effectiveness of the proposed method on this dataset. I understand that the time constraints might have hindered the conduct of more extensive experiments. However, it remains imperative to perform rigorous experiments on large-scale datasets to robustly validate the proposed method's efficiency.
>
> - I believe that the authors can incorporate the new results and modifications as discussed in their response into the revised version of the manuscript. However, given the substantial amount of new content being introduced, there is a legitimate concern regarding the overall quality of the revised version. It is vital to ensure that the revised manuscript maintains the high standards expected for NeurIPS submissions.
>
> Taking all these factors into account, I maintain my perspective that the current iteration of the manuscript does not yet meet the quality threshold expected for acceptance at NeurIPS. Further work addressing the remaining concerns is necessary to elevate the manuscript to the desired standard.

---

> > ### Author Response · Authors · 2023-08-14
> > **Thanks reviewer M5qJ and some clarifications**
> >
> > We thank the reviewer for acknowledging our responses and confirming the resolution of most concerns.
> >
> > Regarding your remaining concerns:
> >
> > - The inductive setting has never been the focus of this work, yet we have shown it is very possible to extend our framework to that setting, which can be studied in depth in a future work. After all, there have been many popular GNN variants that are not inductive in the first place, such as [1, 2, 3].
> >
> > - For MPNNs, effectiveness and scalability are often two pretty orthogonal factors (because MPNNs only work with local graph neighborhoods). Since we are talking about scalability here, we believe the point is to show that our method does scale to larger graph datasets, and the extra experiment we conducted on OGB-Arxiv is already solid proof of this. While we do plan to provide a more comprehensive analysis on OGB in the revision, we don't think missing a full experiment here can significantly hinder the justification of our scalability.
> >
> > - In fact, there are not many new contents that need to be merged into the main paper. All of the new results we added in the PDF are supplementary in nature and can be easily re-organized and put in the Appendix. As for our rebuttal to the reviewers, for each comment that does not only need a clarification response, we have clearly articulated in the response regarding how to improve the paper based on the particular comment.
> >
> > Again we appreciate the reviewer's time and critical comments, but we believe what makes a paper sharable should mostly lie in its being innovative and inspiring, instead of being perfect in every (minor) perspective, even for a high-standard venue like NeurIPS.
> >
> > **References**
> > [1] Perozzi, Bryan, Rami Al-Rfou, and Steven Skiena. "Deepwalk: Online learning of social representations." In Proceedings of the 20th ACM SIGKDD international conference on Knowledge discovery and data mining. 2014.
> > [2] Gasteiger, Johannes, Aleksandar Bojchevski, and Stephan Günnemann. "Predict then Propagate: Graph Neural Networks meet Personalized PageRank." In International Conference on Learning Representations. 2018.
> > [3] Chien, Eli, Jianhao Peng, Pan Li, and Olgica Milenkovic. "Adaptive Universal Generalized PageRank Graph Neural Network." In International Conference on Learning Representations. 2020.

---

### Official Review · Reviewer_y89v · 2023-07-07

**Soundness:** 3 good
**Presentation:** 3 good
**Contribution:** 3 good
**Rating:** 7
**Confidence:** 4

**Summary:**

This paper focuses on the alleviation of overfitting and over-smoothing of deeper GNNs. It is an interesting topic and the solution seems promising with a sample-based node level residual block. Extensive experiments on public datasets verifies the applicability.


**Strengths:**

The paper analyses in a new perspective on the performance drop of GNN with the increase of layers, with motivating examples. It proposes a new residual block with the parameters randomly sampled from a natural distribution. The block alleviates the performance drop and seems applicable in general to all GNN models.

**Weaknesses:**

1. The presentation is in great need to improve. There are some long sentences that hinder the understanding of the authors's ideas, such as L54. A research article should describe facts with a 3rd party stand, but not subjective notations.
2. The paper should be self-contained even without the appendix, but it does not provide enough explanations such as on L83.
3. The figure is not reflecting exactly what is written such as L90 for figure 1, nothing shows the overlaps of aggregations via the lines and nodes.
4. Abbreviations should come with the full spellings on the first occurrence, even if it may be obvious in a specific research domain, such as GCN, but not GCNII on L194.
5. The citations should go to clear items such as a formula or a reference, but not a long section as L202 to Sec. 4.
6. It seems to be unnecessary to have Sec. 3.1 if there is only 1 sub-section.
7. Theorem 1 does not mean anything to me as it gets only formulas, but no conditions neither a conclusion.
8. It should be consistent to have the name of the proposed method or model.

**Questions:**

See weakness

---

> ### Author Rebuttal · Authors · 2023-08-07
>
> Thank you very much for your valuable feedback. Below are our responses to your questions/concerns.
>
> > The presentation is in great need to improve. There are some long sentences that hinder the understanding of the authors’s ideas, such as L54. A research article should describe facts with a 3rd party stand, but not subjective notations.
> >
>
> Thank you for the great suggestion. We will ask the senior authors of this paper to help more with professional writing and perform more thorough polish of the paper. For example, we will change the sentence at L54 to: “From the perspective of k-hop subgraph overlap, we provide a reinterpretation of the phenomenon where the efficacy of traditional message-passing GNNs decreases with an increase in the number of layers.”
>
> > The paper should be self-contained even without the appendix, but it does not provide enough explanations such as on L83.
> >
>
> We will reorganize the paper such as to include necessary explanations for L83 in Section 2.3.
>
> > The figure is not reflecting exactly what is written such as L90 for figure 1, nothing shows the overlaps of aggregations via the lines and nodes.
> >
>
> Thanks for the helpful suggestion. We will redraw the diagram to visualize the overlap in the revision.
>
> > Abbreviations should come with the full spellings on the first occurrence, even if it may be obvious in a specific research domain, such as GCN, but not GCNII on L194.
> >
>
> Thanks for pointing out this issue. We will add full spellings to all abbreviations in the paper upon their first occurrence.
>
> > The citations should go to clear items such as a formula or a reference, but not a long section as L202 to Sec. 4.
> >
>
> We will fix this in the revision.
>
> > It seems to be unnecessary to have Sec. 3.1 if there is only 1 sub-section.
> >
>
> We will remove the redundant subsection title of 3.1.
>
> > Theorem 1 does not mean anything to me as it gets only formulas, but no conditions neither a conclusion.
> >
>
> We acknowledge and agree with your point. The original intention of this paragraph is to give the general term formula of SNR-GCN, which is really not enough to exist as a theorem. We will present the equations directly in the paper instead of providing them in the form of a theorem.
>
> > It should be consistent to have the name of the proposed method or model.
> >
>
> Thank you for bringing this to our attention. We will ensure consistency in the name of the proposed method or model in the revision.

---

> > ### Comment · Reviewer_y89v · 2023-08-18
> >
> > Thank you for the response, I keep my rating unchanged.

---

> > > ### Author Response · Authors · 2023-08-18
> > > **Thanks for the reply**
> > >
> > > We thank the reviewer for the reply and we will properly incorporate the discussions in the rebuttal into our revised paper.

---

### Author Rebuttal · Authors · 2023-08-08

We thank the reviewers for their extensive reviews, helpful comments and actionable suggestions for improving our manuscript. We have made a PDF that includes tables and figures of our extended experimental results. Please refer to our detailed rebuttal for interpreting the results.

---

### Comment · Area_Chair_ixaq · 2023-08-20
**Comparison to subgraph aggregation literature**

Dear authors,

I am worried that you write a paper about subgraph aggregation methods yet fail to even cite prior and recent work on aggregating subgraphs.  I understand that your perspective is that of analyzing GNN architectures with residual connections, but I find this perspective too narrow-minded and after reading the paper in detail, believe that you need to at least place your work in the context of results from the subgraph aggregation literature. Here are a few pointers:

https://arxiv.org/abs/2110.02910
https://arxiv.org/abs/2206.11168

Related to this, I also miss you placing your work in the context of recent literature on oversmoothing and oversquashing. Again, your citations and comparisons seem too self-referential to a niche literature within the GNN community.

Could you please comment on this?

---

> ### Author Response · Authors · 2023-08-20
> **Authors' comments to AC ixap on related literature of subgraph aggregation**
>
> We thank the AC for reading our paper and raising the very valid question about subgraph aggregation. Here we differentiate the concept of *subgraph* in our consideration and the referred studies on *subgraph aggregation* in the following two aspects.
>
> - Context and definition: The referred studies on subgraph aggregation leverage subgraphs of the whole graphs to enhance the *expressiveness of GNNs for graph classification tasks*, so subgraphs there are certain *informative substructures in the whole graph*. In our work, we focus on subgraphs around each node to alleviate the *oversmoothing/overfitting of GNNs for node classification tasks*, so subgraphs here are in fact the *deep neighborhoods of nodes*.
>
> - Challenges and techniques: The referred studies focus on the effective selection process of informative subgraphs and the provable expressiveness of GNNs, while we focus on the proper representation of node neighborhoods and empirical ability of deepening the node neighborhoods. There is also a seemingly related sampling process in both studies. However, the sampling process in the referred studies is mainly for efficiency purposes, but that in our study is used for bringing in randomness to further reduce oversmoothing/overfitting.
>
> We believe our work, except for borrowing the term *subgraph* (without sufficient explanation), is rather different from the referred studies. However, we are eager to include discussions as such into the revision, as well as adding more references and discussions on other existing work on oversmoothing and oversquashing. We thank the AC again for providing this opportunity for us to further justify and improve the position of our work and broaden our discussion on existing literature.
>
> Please let us know if there are any more concerns!

---

> > ### Comment · Area_Chair_ixaq · 2023-08-21
> > **Thanks**
> >
> > Thanks for the answer. I will take it into account when making the recommendation.

---

> > > ### Author Response · Authors · 2023-08-21
> > >
> > > Certainly. We thank the AC for the extended interest and attention, as well as the additional opportunity for broadening our discussion of the literature.

---

### Decision · Program_Chairs · 2023-09-21

**Decision:**

Reject

**Comment:**

There are several weaknesses that warrant in my opinion to reject the paper at this point in time. Since two reviewers scored the paper higher and argued for acceptance, I read and reviewed the paper. After reading the paper, studying related papers, and considering the arguments of all reviewers, I recommend the paper to be rejected. The review of the more negative (and thorough) reviewer and my own assessment weighed more than the (rather shallow and short) reviews of the more positive reviewers.

Here is the justification for my decision:

The main observation and motivation of the paper has been observed and used multiple times before. There are numerous papers analyzing oversmoothing and it is common knowledge that oversmoothing is based on the distance of latent node representations (embeddings)  converging to zero (getting smaller) with increasing depth of the GNN. There exist a variety of different approaches to quantify over-smoothing in GNNs such as measures based on the Dirichlet energy on graphs ([1,2,3]), as well as measures based on the mean-average distance (MAD). While these measures take the feature vectors of all nodes into account, they essentially capture the same mathematical notion of node similarities as the measure (SMV) used in the paper.

With all these measures, oversmoothing is then defined as the latent nodes’ features converging to the same values. Relating these measures also to the observations made in the proposed paper show that the authors’ observation is actually common knowledge: the more the k-hop neighborhoods overlap (with increasing k), the more similar will the latent node features become, the more the measures will converge to 0, and the more oversmoothing will take place. Hence, none of these observation are significant insights and advances over prior work.

[1] Chen Cai and Yusu Wang, A Note on Over-Smoothing for Graph Neural Networks

[2]  Lingxiao Zhao and Leman Akoglu, PairNorm: Tackling Oversmoothing in GNNs

[3] Kaixiong Zhou et al., Dirichlet energy constrained learning for deep graph neural network


Section 3.1 of the paper shows that prior residual GNNs can be formulated as aggregating different powers of the normalized Laplacian. This is what the authors refer to as subgraphs -- for a node, its 1-hop neighborhood is a subgraph, all nodes exactly 2 hops away is a subgraph etc (Later I will argue why I believe this is problematic as a term.) These subgraphs are aggregated. While this is interesting to make this aggregation of different powers of the Laplacian explicit, that residual connections result in this is entirely unsurprising due to the (recursive) definition of GNNs.

Hence, the contribution “We reinterpret the phenomenon that the effectiveness of traditional message-passing GNNs decreases as the number of layers increases from the perspective of k-hop subgraph overlap” is not new and common knowledge. Moreover, the second claimed contribution is also not surprising or particularly insightful. With all due respect, claiming that making this observation explicit is a “theoretical analysis” is an overclaim. After some criticism of a reviewer, the authors also agreed that Theorem 1 is not really a theorem but a simple rewriting of the initial definition of the proposed method.

The authors now proceed to propose their model. In a nutshell, it learns for each layer a stochastic weight that determines how much of the representation at depth k-1 relative to the representation at layer 1 is added -- in other words, the strength of the residual connections. The weight is sampled indirectly through a learnable Gaussian and backprop accomplished with a simple reparameterization trick. In other words, the authors propose a simple stochastic weight mechanism to determine the residual connections specific to the depth and the nodes. Similar ideas have been proposed in structure learning on a more fine-grained level. Now, not entirely novel and simple is still good if it works well, but the experimental results are not strong at all. Most of the results in Table 3 are not statistically significant and it is not indicated what depth was actually used. Table 4 shows that increasing the number of layers is almost always worse, which shows that having more layers doesn’t make sense for these datasets, even with residual connections.  Table 5 shows a slight improvement over other residual baselines.

While some of the prior work on oversmoothing has been cited, the connection to existing methods (e.g. rewiring methods, structure learning methods) is not made clear and explicit enough. For instance, there is a large body of work on oversmoothing and the authors could have related their methods much better to these existing methods and ways to alleviate overmoothing. A prominent example is the literature on rewiring and subgraph aggregation methods:

https://arxiv.org/abs/2211.15779 (rewiring is essentially a way to directly and in a fine-grained manner change the aggregation of what the authors call subgraphs -- plus references herein, also mitigates oversmoothing)
https://arxiv.org/abs/2110.02910 (ideas to aggregate subgraphs, including ego-subgraphs which are k-hop subgraphs, including follow up work). The authors are very narrowly focused on residual GNNs and a comparison to a few methods from this community.

Finally, the use of term “subgraph aggregation” is problematic due to it being used in a large and growing body of work on subgraph aggregation methods. Here, the subgraphs are determined by heuristics (and combined) or determined in a data-driven way. Even if this research area is distant enough from the proposed method and residual GNNs (which I disagree with) it is important to make clear upfront that the term is used differently. In this case it might also be problematic to use the term in the title of the paper.